# Selective sorting of hexane isomers by anion-functionalized metal-organic frameworks with optimal energy regulation

Qingju Wang [1,2], Lifeng Yang [1], Tian Ke[1], Jianbo Hu [1], Xian Suo [2], Xili Cui [1,2,3] ✉ & Huabin Xing [1,2] ✉

Extensive efforts have been made to improve the separation selectivity of hydrocarbon isomers with nearly distinguishable boiling points; however, how to balance the high regeneration energy consumption remains a daunting challenge. Here we describe the efficient separation of hexane isomers by constructing and exploiting the rotational freedom of organic linkers and inorganic $SnF_6^{2-}$ anions within adaptive frameworks, and reveal the nature of flexible host-guest interactions to maximize the gas-framework interactions while achieving potential energy storage. This approach enables the discrimination of hexane isomers according to the degree of branching along with high capacity and record mono-/di-branched selectivity (6.97), di-branched isomers selectivity (22.16), and upgrades the gasoline to a maximum RON (Research Octane Number) of 105. Benefitting from the energy regulation of the flexible pore space, the material can be easily regenerated only through a simple vacuum treatment for 15 minutes at 25 °C with no temperature fluctuation, saving almost 45% energy compared to the commercialized zeolite 5 A. This approach could potentially revolutionize the whole scenario of alkane isomer separation processes.

The world's pursuit of maximum energy efficiency drives the petrochemical industry to optimize the separation processes which consume about 10–15% of the global energy consumption[1]. Among these processes, the adsorptive separation of multi-component alkane isomers with inert surfaces, subtle molecular size difference and similar polarizabilities represents a critical and challenging one[2–6]. The widely used adsorbent of molecular sieve zeolite 5A suffers from two drawbacks[7,8]: (i) zeolite 5A is unable to distinguish mono-branched and di-branched alkane isomers, limiting the RON upgrade of gasoline; (ii) the relatively rigid backbone, narrow pore and strong cation binding sites of zeolite 5A cause high energy cost for regeneration, which usually requires the heating temperature over 473 K.

To overcome these drawbacks, researchers have been working on screening and exploring novel adsorbents[9–15]. Until now, limited success has been achieved in this area. Hexane ($C_6H_{14}$) isomers are important components of gasoline (~20%)[2], with the global annual consumption up to 9500 million barrels[16]. Notably, the di-branched isomers, 2,3-dimethylbutane (23DMB) and 2,2-dimethylbutane (22DMB), exhibit the high RONs of 105 and 94, respectively, whereas the linear n-hexane (nHEX), shows low RONs of 30. The mono-branched 2-methylpentene (2MP) and 3-methylpentene (3MP) have moderate RONs of 74 and 75, respectively. In the case of separating hexane ($C_6H_{14}$) isomers for high-quality gasoline with higher RONs, pioneer studies showed that triangular pore geometry of $Fe_2(BDP)_3$ framework realized the thermodynamic separation of hexane isomers by the difference of the isomers to wedge along the triangular corners[5]. However, all five hexane isomers could enter the pore, and the lack of selective binding sites resulted in the weak separation selectivity. Most

[1]Key Laboratory of Biomass Chemical Engineering of Ministry of Education, College of Chemical and Biological Engineering, Zhejiang University, Hangzhou, China. [2]Hangzhou Global Scientific and Technological Innovation Center, Zhejiang University, Hangzhou, China. [3]Shanxi-Zheda Institute of Advanced Materials and Chemical Engineering, Hangzhou, China. ✉e-mail: cuixl@zju.edu.cn; xinghb@zju.edu.cn

of the current research focused on the separation of easy-handled three-component mixtures (nHEX/3MP/22DMB) with larger size differences[4,17–19], and limited works addressed the realistic separation condition of more complex five-component mixtures (nHEX/2MP/3MP/23DMB/22DMB)[5,20,21], like Al-bttotb[20], but showed relatively low selectivity for mono- and di-branched alkane isomers. In addition, within these five components, 23DMB has the highest RON (105) and needs further purification to obtain the highest quality gasoline, meeting more application scenarios. Furthermore, the research on the regeneration energy of adsorbents in alkane isomers separation processes has long been underappreciated, which is crucial for practical applications.

An ideal efficient separation process should possess high capacity and selectivity as well as low regeneration energy consumption. However, these objectives often conflict because of the trade-off between high affinity and low heat of desorption. Recently, there are a number of flexible MOFs reported to have inherent thermal management applied to achieve highly efficient separation/storage of gases[22–28]. These materials can reversibly utilize latent heat during the phase change process for thermal energy storage and release, enabling efficient adsorption–desorption cycles. Here, we put forward an approach that constructing and exploiting the rotational freedom of organic linkers and inorganic $SnF_6^{2-}$ anions within adaptive frameworks, to achieve the highly efficient separation of alkane isomers (Fig. 1a). The anion-functionalized ZU-72 (also termed SNFSIX-2-Cu-i, SNFIX = $SnF_6^{2-}$, 2 = 4,4'-dipyridylacetylene, i = interpenetrated) shows extraordinary discrimination for hexane isomers, adsorbing linear and mono-branched alkanes with high capacity, and excluding the di-branched isomers, a record selectivity of mono-/di-branched isomers (6.97). Meanwhile, the reversible phase transition of ZU-72 triggered by guest molecules further enables the rational regulation of energy during adsorption/desorption processes (Fig. 1b), which contributes to its mild regeneration condition (298 K). More importantly, to obtain the highest quality gasoline with RON of 105, SIFSIX-1-Cu (SIFIX = $SiF_6^{2-}$, 1 = 4,4'-bipyridine) with suitable pore size, are testified as an excellent sieve for the separation of 22DMB and 23DMB. The dynamic

breakthrough experiments confirm the unprecedented separation performance.

## Results

### Structural characterization

Previous works have shown that $SiF_6^{2-}$ anions showed moderate binding with H atoms of light hydrocarbons[29–31]. Taking advantage of the anions, further advances would be realized by tuning the spatial arrangements of the selective binding sites according to the shape of hexane isomers. Thus, the interpenetrated anion-functionalized MOFs ZU-72 (ZU = Zhejiang University, also termed SNFSIX-2-Cu-i, 2 = 4,4'-dipyridylacetylene, i = interpenetrated), isomorphic to the SIFSIX-2-Cu-i, with pore sizes distributed around 3.6–5.9 Å, are synthesized. The three-dimensional frameworks are formed by six-copper node coordinated with four organic linkers and then pillared with $SnF_6^{2-}$ anions (Supplementary Fig. 1a). The anions create a strong electrostatic environment to trap guest molecules (Supplementary Fig. 1b). As shown in Supplementary Fig. 1c, through interpenetration, the anions are orderly arranged to the "zigzag" geometry, which can recognize isomers with different shapes. Nitrogen adsorption at 77 K on ZU-72 gives a BET surface area of 572 m² g⁻¹ (Supplementary Fig. 5). Thermogravimetric analysis reveals that ZU-72 is stable up to 496 K (Supplementary Fig. 6). The non-interpenetrated SIFSIX-1-Cu (1 = 4,4'-dipyridine) is constructed by six-copper node coordinated with four organic linkers and then pillared with $SiF_6^{2-}$ anions, with pore sizes distributed around 6.8–7.2 Å (Supplementary Fig. 2), larger than that of ZU-72. ZU-61 (also termed NbOFFIVE-1-Ni) is isomorphic to SIFSIX-1-Cu, with slightly wider pore sizes distributed around 7.3–7.8 Å (Supplementary Fig. 4). The bulk purity of these anion-functionalized MOF samples is confirmed by the powder X-ray diffraction (PXRD) tests (Supplementary Figs. 7–11).

### Hexane isomers separation performance

The single-component equilibrium isotherms reveal that ZU-72 shows prominent discrimination for hexane isomers according to the degree of branching. As shown in Fig. 2a, the uptake of nHEX on ZU-72 is up to

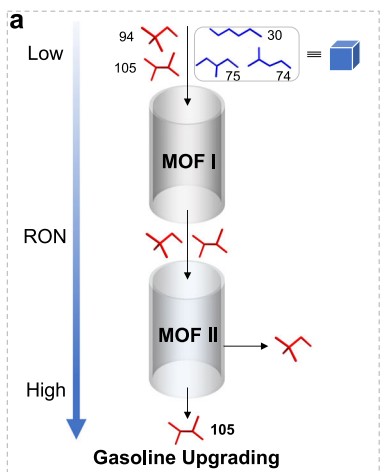

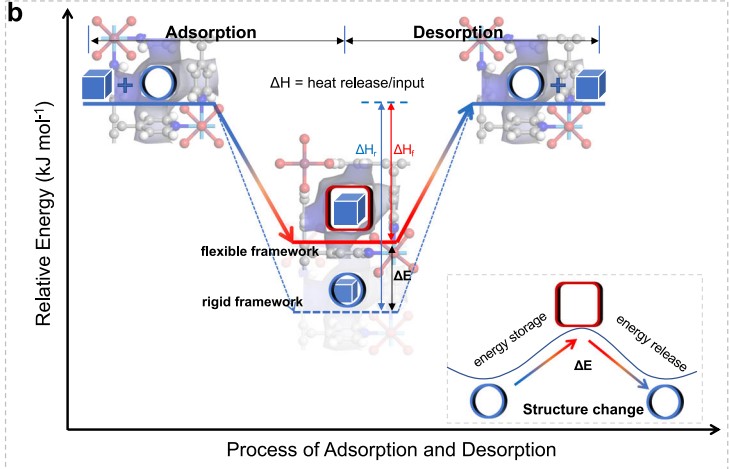

**Fig. 1 | Schematic diagrams of the separation performance for hexane isomers and the thermal management of anion-functionalized MOFs. a** Firstly, the MOF I (ZU-72) can screen out di-branched isomers from the other three and augment the RON of gasoline. Then, the MOF II (SIFSIX-1-Cu) is an excellent sieve for separating the most valuable 23DMB from di-branched isomers, further upgrading the final product to a maximum RON of 105. **b** Energy profiles of adsorption/desorption processes for flexible and hypothetical rigid frameworks of ZU-72, where the sky-blue cube represents guest molecules, the skyblue cylinder and vermilion square box represent the pore structures of pristine and guest-loaded host framework

(ZU-72), respectively; and $\Delta H$ is the adsorption enthalpy, $\Delta E$ is the energy consumed for the structural transformation of the host framework. When the host framework is rigid, the host–guest binding energy is the adsorption enthalpy, i.e., $\Delta H = \Delta H_r$. However, there exists inherent thermal management in the flexible ZU-72, where the host–guest binding energy is partially assigned for the deformation of the host framework and stored in it ($\Delta E$), following the latent energy stored in the backbone will be resupplied for regeneration during the desorption process. The illustration is a schematic diagram of the energy transfer corresponding to the structural changes of the host framework during adsorption/desorption processes.

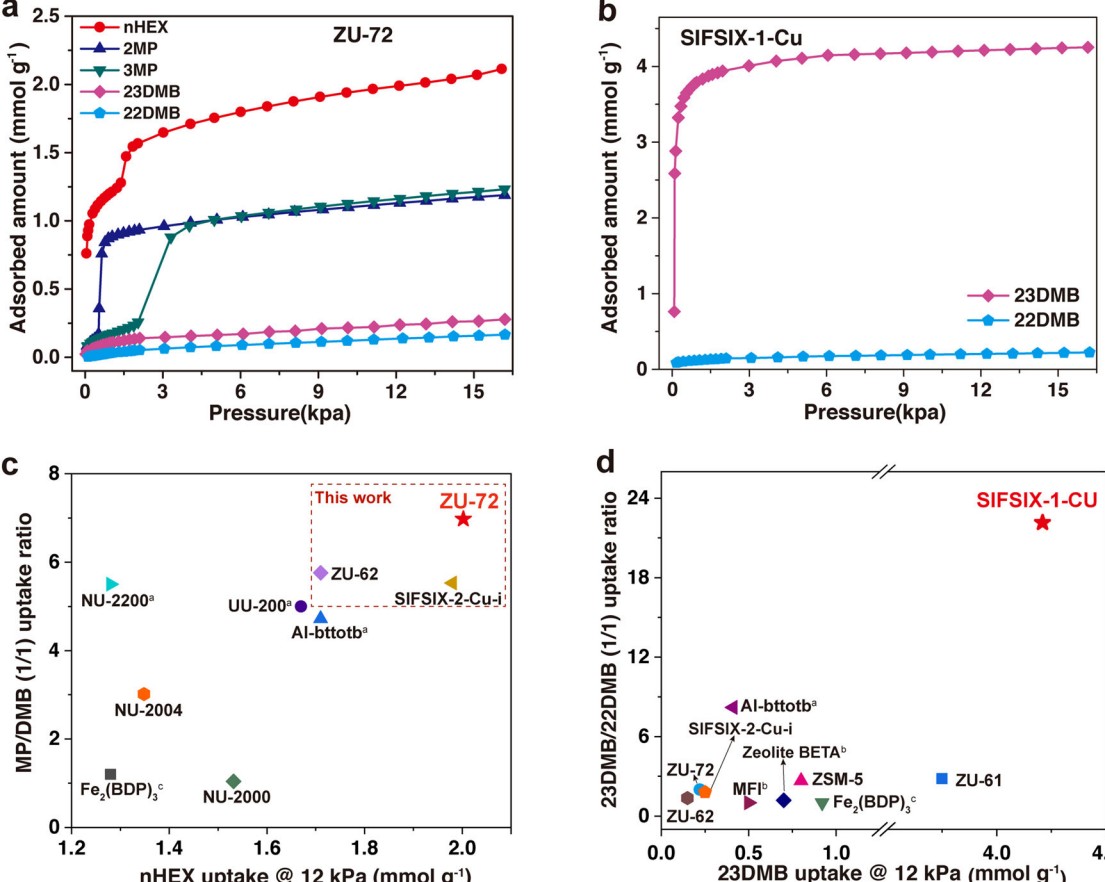

**Fig. 2 | Hexane isomers adsorption isotherms of anion-functionalized MOFs.** Pure-component vapor adsorption isotherms of nHEX, 2MP, 3MP, 23DMB, and 22DMB on ZU-72 at 298 K in the region of 0–16 kPa (**a**). Pure-component vapor adsorption isotherms of 23DMB and 22DMB on SIFSIX-1-Cu at 298 K in the region of 0–16 kPa (**b**). Plots of the nHEX uptake as a function of the MP/DMB uptake ratio for ZU-72 and other previously reported materials at 298 K and 12 kPa (**c**). **d** Plots of the 23DMB uptake at 12 kPa as a function of the 23DMB/22DMB uptake ratio for SIFSIX-1-Cu and other previously reported materials at 298 K. [a] at 303 K, [b] at 423 K and [c] at 433 K. MP/DMB is defined as (2MP + 3MP)/(23DMB + 22DMB).

2.12 mmol g$^{-1}$ at 16 kPa and 298 K, which is 1.31 times higher than that of zeolite 5 A (1.62 mmol g$^{-1}$, Supplementary Fig. 16) and higher than other good performing MOFs (Fig. 2c). Meanwhile, ZU-72 can distinguish mono-branched isomers from the linear nHEX with the uptake of 1.19 mmol g$^{-1}$ for 2MP and 1.23 mmol g$^{-1}$ for 3MP. The di-branched isomers, 23DMB and 22DMB, are not allowed to be adsorbed on ZU-72 owing to their larger molecular size (5.6 Å and 6.2 Å, respectively), and the complete size sieving effect is also confirmed by the dynamic adsorption test (Supplementary Fig. 12), endowing ZU-72 with the record separation selectivity of mono-/di-branched (6.97), higher than the isostructural SIFSIX-2-Cu-i (5.53) and previous benchmark materials, Fe$_2$(BDP)$_3$ (1.21)[5] and Al-bttotb (4.72)[20] (Fig. 2c and Supplementary Fig. 21). This is a remarkable advancement since it is a great challenge to precisely cut hexane isomers up to five components based on the degree of branching, especially with the ability to exclude both two di-branched isomers from the mono-branched ones, which has rarely been achieved[15]. The distinctive attributes of ZU-72 is the introduction of SnF$_6$$^{2-}$ anion compared with other anion-functionalized isostructures (Supplementary Fig. 38)[32–34]. Furthermore, for the larger di-branched hexanes, SIFSIX-1-Cu with suitable pore size and large crystal size realizes the size exclusion effect of 22DMB (8.0 × 6.7 × 5.9 Å$^3$) from 23DMB (7.8 × 6.7 × 5.3 Å$^3$), with the remarkably high capacity of 23DMB (4.25 mmol g$^{-1}$ at 16 kPa and 298 K) and record selectivity of di-branched isomers (22.16) (Fig. 2b, d). The sieving performance is also evidenced by the differential scanning calorimetry experiments (Supplementary Fig. 18) and the dynamic adsorption test (Supplementary Fig. 19). It is worth mentioning that for the separation of gases with

extremely similar structures, achieving such a high adsorption capacity under the circumstance of complete size exclusion is attractive. In addition, for the currently popular separation (nHEX/3MP/22DMB), SIFSIX-1-Cu also shows the best separation performance than previously reported benchmarks (Supplementary Figs. 20 and 22). We also observe the efficient separation performance of 23DMB and 22DMB in ZU-61 (Supplementary Fig. 24). To the best of our knowledge, the exceptional performance is unprecedented and enables the possible production of gasoline with RON higher than 100.

Furthermore, the breakthrough experiments were conducted to demonstrate the actual separation ability of the above materials (see Supplementary Information for breakthrough measurement details). As the equimolar quinary mixture of nHEX, 2MP, 3MP, 23DMB, and 22DMB passed through the column packed with activated ZU-72, a separated eluted sequence was observed for alkanes with different branching. As shown in Fig. 3a, the di-branched 23DMB and 22DMB, as the most desirable gasoline components with high RONs, broke through almost immediately, followed by the mono-branched 2MP and 3MP (~50 mL), between which the RON of the eluted product was higher than 98 as calculated from the breakthrough curves, significantly higher than the value of 87 using the zeolite 5A (Supplementary Fig. 27). The nHEX exhibited the longest retention time of 230 mL, and the working capacity for ZU-72 was 1.69 mmol g$^{-1}$, exceeding the previous benchmark material, Fe$_2$(BDP)$_3$ (0.41 mmol g$^{-1}$). The productivity of high-quality gasoline (RON > 95) for ZU-72 (15.8 mL g$^{-1}$) was higher than that of SIFSIX-2-Cu-i (14.3 mL g$^{-1}$), Fe$_2$(BDP)$_3$ (1.5 mL g$^{-1}$) (Supplementary Fig. 32).

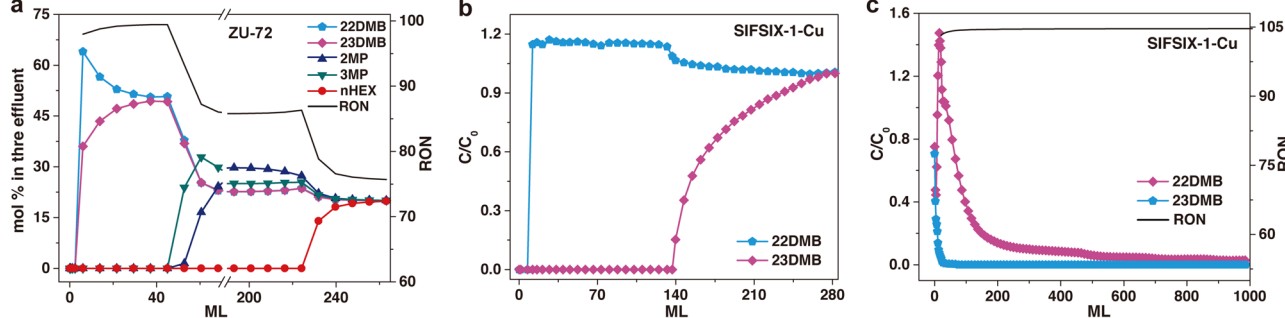

**Fig. 3 | Breakthrough curves of anion-functionalized MOFs. a** Breakthrough experiments for the equimolar mixture of nHEX/2MP/3MP/23DMB/22DMB on ZU-72 together with the RON calculated form the eluted mixture at 298 K. **b** Breakthrough experiments for the equimolar mixture of 23DMB/23DMB on SIFSIX-1-Cu at 298 K. **c** The desorption curves of 23DMB and 22DMB during the regeneration process under the nitrogen flow together with the RON calculated form the eluted mixture on SIFSIX-1-Cu. The horizontal axis represents the volume of the mixture of nitrogen gas and hexane isomers.

Moreover, we also measured the 23DMB/22DMB binary mixture performance on SIFSIX-1-Cu and ZU-61, a clean and sharp separation of the two di-branched hexanes was observed (Fig. 3b and Supplementary Fig. 34). The 22DMB eluted immediately with a negligible uptake of 0.05 mmol g⁻¹ on SIFSIX-1-Cu due to the exclusion effect, while the 23DMB was trapped in the column retaining 140 mL (Fig. 3b). The following desorption operations enabled the further upgrading of gasoline with the highest RON (~105) (Fig. 3c and Supplementary Fig. 35), with a productivity of 55.6 mL g⁻¹. In addition, the cycle and regeneration capabilities were further investigated by breakthrough tests (Supplementary Figs. 33 and 36), there is no noticeable decrease of breakthrough times during the three cycles, revealing excellent recyclability. The exceptional separation performances of ZU-72, SIFSIX-1-Cu, and ZU-61 announced their great potential for gasoline upgrading. And the demonstrated strategy is also applicable for the efficient separation of pentane isomers (Supplementary Fig. 14) and heptane isomers (Supplementary Figs. 15 and 23).

## Energy of regeneration

Except for the high capacity and selectivity, the energy consumption for regeneration of porous materials is also vital in alkane isomers separation, which is closely related to the energy efficiency[35–40]. In this regard, ZU-72, designed for hexane isomers separation, takes up 30% more nHEX than commercial zeolite 5A (Fig. 4a). Meanwhile, ZU-72 exhibits an excellent adsorption rate of nHEX, better than zeolite 5A (Fig. 4b), which is essentially important to improve the working capacity as well as the regeneration efficiency. Also, ZU-72 is easy to be regenerated under moderate regeneration condition only through a simple vacuum processing without additional heating (~25 °C), just for 15 min (Fig. 4a and Supplementary Fig. 13), and the invariable cyclic adsorption–desorption performance verifies its easy regeneration performance (Fig. 4c). In contrast, much higher temperatures, up to 200 °C for 8 h, are the prerequisites for zeolite 5A to realize the completed regeneration (Fig. 4a and Supplementary Fig. 17). The differential scanning calorimetry (DSC) experiments are further conducted to measure the heat released of nHEX on ZU-72 and zeolite 5A, quantitatively evaluating the corresponding regeneration energy. About 47 kJ of energy input is sufficient to regenerate 1 mol of nHEX adsorbed on ZU-72, saving almost 45% of energy compared to the energy for regeneration of zeolite 5A (86 kJ mol⁻¹) under the same conditions (Fig. 4d).

The large energy saving of ZU-72 attracted us to further explore the latent energy management during the sorption processes. The single-crystal X-ray diffraction (SCXRD) measurements were performed on the bare (activated) and nHEX-loaded ZU-72. An orientational disorder-order transition of both pyridine rings and $SnF_6^{2-}$ anions was observed after the accommodation of the guest molecules.

The flexibility of the structure accounts for the hysteresis behavior observed in the adsorption isotherms. In the bare ZU-72, the pyridine rings formed a plane angle of ±22.5° with the ac (or bc) crystallographic plane with neighboring rings, and the exposed F atoms exhibited highly disordered (state I) (Fig. 4g), while in the nHEX-loaded MOF structure, the pyridine rings were vertical, parallel to the c axis and the exposed F atoms extended to the 0 or 38° (state II) (Fig. 4h). It is reasonable to believe that the adaptive structural transformation of ZU-72 would impose effects on the energy transition, and its specific role is further explored by the total energy calculation as a function of the rotational angle of $SnF_6^{2-}$ or the pyridine rings using the first-principles DFT-D (dispersion-corrected density functional theory) method. As shown in Fig. 4i, j, the rotational behavior of either pyridine or $SnF_6^{2-}$ anions would trigger the energy release/storage of the framework, and the energy contribution to the framework was about 9–25 kJ mol⁻¹. Also, the above structural change from state I to state II caused by the load of nHEX seemed to function as the energy storage part within the framework. However, the above calculation neglected the potential correlation between the two rotational behaviors, it was not appropriate to simply add or subtract the energy changes caused by the single factor. Therefore, to verify and reveal the specific value of energy storage, the conformation energy of the two states was determined separately, and the results indicated that the ZU-72 framework required ~15 kJ mol⁻¹ energy input to achieve the structural transformation, from state I to state II. The intrinsic thermal management capability of ZU-72 is relative higher than the Fe(bdp) (8.1 kJ mol⁻¹) and Co(bdp) (7.0 kJ mol⁻¹) utilized for $CH_4$ storage[22]. Meanwhile, such a structural transformation is fully reversible (Supplementary Fig. 37), that is, the potential energy stored in the backbone due to the structural change upon adsorption will be delivered during the desorption process (transition from state II to state I), which leads to the moderate regeneration condition. Such an energy offset phenomenon triggered by the adaptive backbone is believed to exert positive influence to lower the energy input required for the regeneration.

In addition, the adsorption behavior of nHEX on ZU-72 and zeolite 5A was determined by the MC simulation and further DFT-D theory optimization. In ZU-72, the nHEX molecule was adsorbed within a triangular electrostatic domain ("zigzag" anion sites) created by the three $SnF_6^{2-}$ anions from different nets, and almost all the hydrogen of nHEX was able to find an available F-binding site. The distance of C–H···F interactions was about 2.10–3.20 Å (Fig. 4e), and the DFT-D-calculated static adsorption energy ($\Delta E$) was 46.08 kJ mol⁻¹. The dense multiple interactions led to its high nHEX capacity. While in the zeolite 5A, except for the dense C–H···O interactions (2.0–3.20 Å) between nHEX and the available on the pore surface, the carbon atoms of nHEX exhibited another strong interaction with the free cations (Na⁺, Ca²⁺) inside the pore channel (Fig. 4f), making the nHEX molecule too tightly bound, and the

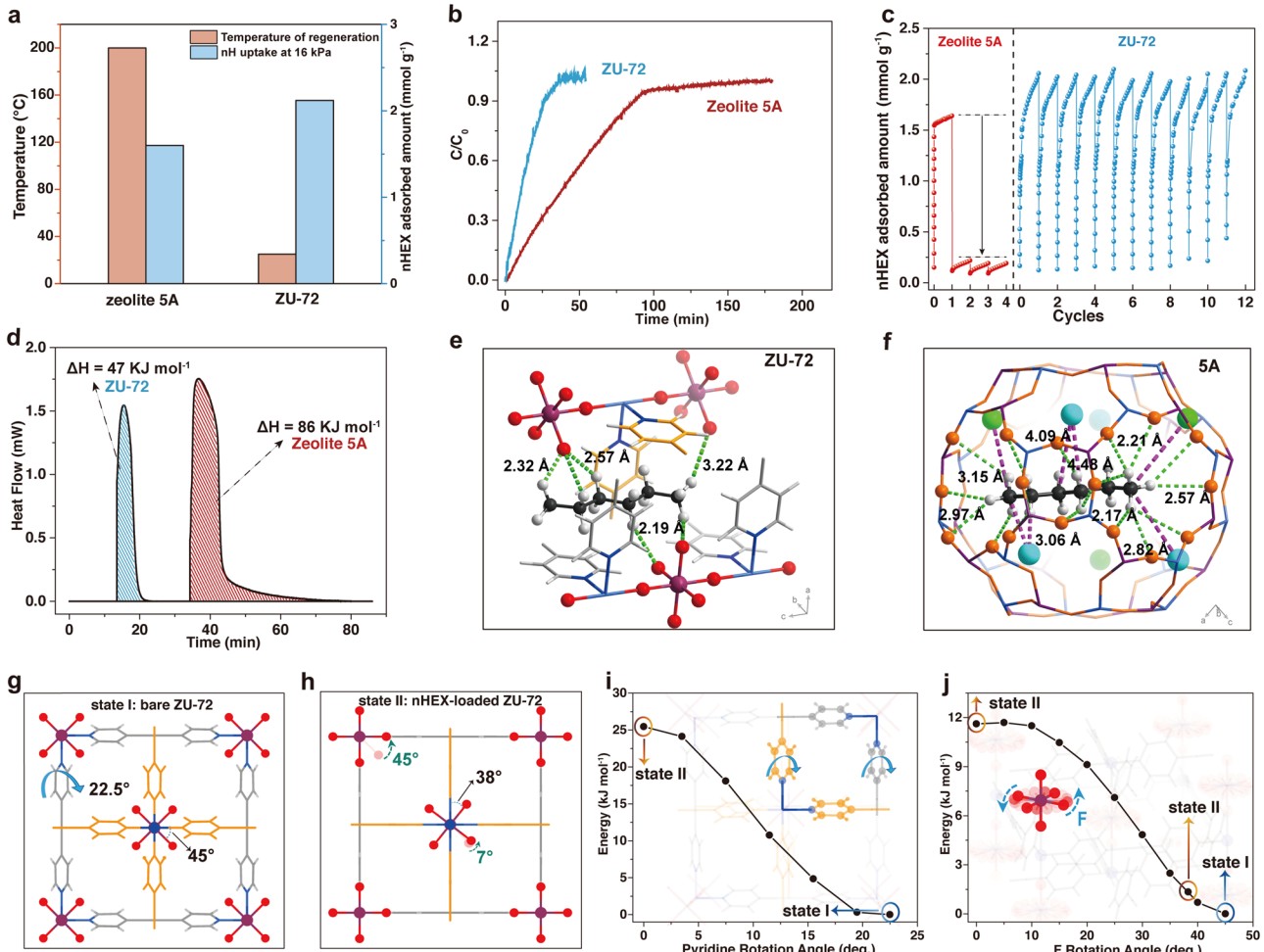

**Fig. 4 | Regeneration energy of anion-functionalized MOFs and commercial zeolite 5A.** Comparison and relation between nHEX uptake at 16 kPa and regeneration temperature of ZU-72 with zeolite 5A (**a**). Time-dependent adsorbed amount profiles of nHEX on ZU-72 with zeolite 5A at 298 K (**b**). The nHEX adsorption–desorption cycles on ZU-72 and zeolite 5A at the regeneration temperature of 298 K (**c**). Calorimetric measurements of nHEX adsorption on ZU-72 and zeolite 5A (**d**). **e, f** The DFT-D calculated binding sites of nHEX within ZU-72 (**e**) and zeolite 5A (**f**). **g, h** Single-crystal structure of bare ZU-72 (**g**), ZU-72·nHEX (**h**). **i, j** The variation of the total energy as the pyridine ring and $SnF_6^{2-}$ anions rotate around the lattice axis $a/b$ and $c$, respectively, derived from DFT-D calculations. Schematic of the rotating units are shown as inset.

corresponding binding energy was as high as 118.29 kJ mol⁻¹. Experience has indicated that the physisorbents with binding energies on the order of 45–60 kJ mol⁻¹ lie in the "sweet spot"[41,42]. Below that, the adsorption capacity cannot be guaranteed, while above that makes excessive regeneration energy consumption. Overall, the intrinsic energy management of the flexible framework and the suitable strength of host–guest interactions achieve the high adsorption capacity and low regeneration energy of ZU-72, making it promising for alkane isomers separation.

## Discussion

The construction and exploitation of the rotational freedom of organic linkers and $SnF_6^{2-}$ anions within adaptive frameworks are efficient for the discrimination of alkane isomers with different degrees of branching. The anion-functionalized ZU-72 realizes the complete size sieving of di-branched from mono-branched isomers with high selectivity and capacity. The framework with some degree of rotational freedom allows the rationally regulation of energy during adsorption and desorption processes, which contributes to the lower energy consumption of regeneration. The successful separation of 22DMB and 23DMB by SIFSIX-1-Cu enables the further upgrading of gasoline to a maximum value (105). It should also be noted that to achieve efficient

separations under practical operating conditions, the granulation of these materials and economic energy consumption assessments need to be considered, and we have already started to address these issues. Overall, this work offers a great advance toward gasoline upgrading, and also emphasizes the contribution of the responsive flexible skeleton, the intrinsic energy regulation makes the separation process be more efficient.

## Methods

### Preparation of ZU-72 (Cu(4,4′-bipyridylacetylene)$_2$SnF$_6$)

The single crystals of ZU-72 were synthesized by slow diffusion of a methanol solution (4.0 mL) of 4,4′-bipyridylacetylene (0.017 g, 0.1 mmol) into an aqueous solution (4.0 mL) of $(NH_4)_2SnF_6$ (0.013 g, 0.05 mmol) and Cu(BF$_4$)$_2$·xH$_2$O (0.012 g, 0.05 mmol) after 1 week.

**Powder synthetic method.** A methanol solution (4.0 mL) of 4,4′-bipyridylacetylene (0.055 g, 0.286 mmol) was mixed with an aqueous solution (4.0 mL) of $(NH_4)_2SnF_6$ (0.069 g, 0.26 mmol) and Cu(BF$_4$)$_2$·xH$_2$O (0.061 g, 0.26 mmol). Then, the mixture was heated at 353 K for 24 h. The obtained powder was filtered, washed with methanol, and exchanged with methanol for 1 days. The yield of ZU-72 is about 70%.

## Preparation of SIFSIX-1-Cu (Cu(4,4'-bipyridine)₂SiF₆·8H₂O)n

Overall, 0.35 g 4,4'-bipyridine was dissolved in 40 mL ethylene glycol at 343 K and an aqueous solution 20 mL of Cu(BF₄)₂·xH₂O (0.266 g) and (NH₄)₂SiF₆ (0.199 g) was added to the former solution. The mixture was then heated at 343 K for 3 h with stirring. The obtained purple powder was washed with methanol, and exchanged with methanol for 1 day[31]. The yield of SIFSIX-1-Cu is about 77%.

## Preparation of ZU-61 (Ni(4,4'-bipyridine)₂ NbOF₅)n

In all, 0.35 g 4,4'-bipyridine was dissolved in 40 mL ethylene glycol at 338 K and an aqueous solution 20 mL of NiNbOF₅ (0.35 g) was added to the former solution. The mixture was then heated at 338 K for 3 h with stirring. The obtained powder was washed with methanol, and exchanged with methanol for 1 day[43]. The yield of ZU-61 is about 65%.

## Preparation of nHEX-loaded ZU-72 crystals

The obtained single crystals of ZU-72 were filtered, washed with methanol, and exchanged with methanol for 2 days. Next, they were degassed at 298 K until the pressure dropped below 10 μmHg. The activated single crystals were loaded into the sample tube for adsorption test, the inner wall of the sample tube was coated with oil (polybutenes), which slowly flowed down the tube wall. Then, the nHEX vapor was backfilled. After ZU-72 single crystals were sealed by oil, they were taken out for the single-crystal X-ray diffraction tests.

## Single-crystal X-ray diffraction

Single-crystal X-ray diffraction data for ZU-72 were collected on a Bruker D8 VENTURE diffractometer equipped with a PHOTONII/CMOS detector (GaK$_\alpha$, $\lambda = 1.314139$ Å)[44]. Indexing was performed using APEX3. Data integration and reduction were completed using SaintPlus 6.01. Absorption correction was performed by the multi-scan method implemented in SADABS. The space group was determined using XPREP implemented in APEX3. The structure was solved with SHELXS-2018 (direct methods) and refined on F2(nonlinear least-squares method) with SHELXL-2018 contained in APEX3 program packages. All non-hydrogen atoms were refined anisotropically.

## Powder X-ray diffraction

Powder X-ray diffraction (PXRD) data were collected using a SHIMADZU XRD-600 diffractometer (Cu K$_\alpha$, $\lambda = 1.540598$ Å) with an operating power of 40 KV, 40 mA, and a scan speed of 4.0° min⁻¹. The data were collected in the range of $2\theta = 5$–40°[44].

## Gas adsorption measurements

ZU-72, SIFSIX-1-Cu, ZU-61 and zeolite 5A were evacuated for 12 h at 353 K, 338 K, and 473 K, respectively, until the pressure dropped below 7 μmHg. Nitrogen adsorption–desorption isotherms at 77 K were collected using ASAP 2460 Analyzer (Micromeritics). The single-component vapor adsorption isotherms of hexane isomers were collected at 298 K on activated samples using autosorb iQ/ASiQwin (Quantachrome Instruments) vapor adsorption analyzer. The single-component vapor adsorption isotherms of pentane and heptane isomers were collected on the BELsorp max II.

## Breakthrough experiments

The breakthrough experiments were carried out in dynamic gas breakthrough equipment. All experiments were conducted using a stainless steel column (4.6 mm inner diameter × 50 mm). According to the different particle size and density of the sample powder, the weight packed in the column was: 0.3319 g for ZU-72, 0.2734 g for SIFSIX-1-Cu, 0.2886 g for ZU-61 and 0.38 g for zeolite 5A, respectively. The column packed with adsorbent was first purged with nitrogen flow at room temperature. The mixed gases of nHEX/2MP/

3MP/23DMB/22DMB (1/1/1/1/1), nHEX/2MP/22DMB (1/1/1) and 23DMB/22DMB (1/1) in nitrogen were produced by nitrogen-blow bubbled method. A nitrogen flow at a set rate of 2 mL/min was bubbled through a mixture of hexane isomers according to the following volumes (the volumes were determined through trial and error and calculated by GC: the experiment was run without any sample and the vapor phase ratios were optimized to an equimolar mixture): 4.212 g of nHEX, 3.014 g of 2MP, 3.604 g of 3MP, 2.72 g of 23DMB and 2.04 g of 22DMB for nHEX/2MP/3MP/23DMB/22DMB quinary mixture (partial pressure of each component is 5.46 kPa); 7.738 g of nHEX, 5.43 g of 2MP, and 3.58 g of 22DMB for nHEX/2MP/22DMB ternary mixture (partial pressure of each component is 7.46 kPa); 5.71 g of nHEX, 4.85 g of 3MP and 2.65 g of 22DMB for nHEX/3MP/22DMB ternary mixture (partial pressure of each component is 6.98 kpa); 8.59 g of 23DMB and 6.14 g of 22DMB for 23DMB/22DMB binary mixture (partial pressure of each component is 9.58 kPa). Outlet gas from the column was monitored using gas chromatography (GC-2010, SHIMADZU) with a KB-1MS capillary column. After the breakthrough experiment, the adsorption bed was regenerated by nitrogen flow (1–5 mL min⁻¹) for 5–24 h at 298 K - 473 K. Specifically, the temperatures of regeneration are 298 K for ZU-72, 343 K for SIFSIX-1-Cu and 358 K for ZU-61.

The desorption curves for SIFSIX-1-Cu and ZU-61 were obtained in a two-step process by adsorption followed by desorption. Firstly, the 23DMB/22DMB binary mixture was introduced into columns at 298 K. When the breakthrough experiments were finished, a flow rate of from 1 mL min⁻¹ to 5 mL min⁻¹ N₂ was introduced, and the columns were heated from 298 to 343 K for SIFSIX-1-Cu and 358 K for ZU-61, respectively. Outlet gas from the column was monitored using gas chromatography (GC-2010, SHIMADZU) with a KB-1MS capillary column.

## Kinetic adsorption measurement

The time-dependent adsorption profiels of hexane isomers were measured on the PE TGA. Frist, ZU-72 and zeolite 5A samples were evacuated for 12 h at 353 and 473 K, respectively, until the pressure dropped below 7 μmHg in advance. Then, they were transferred to the PE TGA instrument and activated at certain temperatures under dry nitrogen flow until the weight remained stable. The SIFSIX-1-Cu sample was directly activated on the PE TGA at 343 K under dry nitrogen flow until the weight remained stable. Upon the analysis started, the nitrogen flow was immediately changed to hexane isomers bubbled with nitrogen at a certain flow rate, and the mass of the sample loaded with hexane isomers was continuously recorded.

## Differential scanning calorimetry

The enthalpy of adsorption for hexane isomers were measured using the PE TGA and PE DCS 7 instruments. First, ZU-72 and zeolite 5A samples were evacuated for 12 h at 353 and 473 K, respectively, until the pressure dropped below 7 μmHg in advance. Then, they were wrapped in aluminum crucibles and transferred to the PE TGA instrument. Finally, they were activated at certain temperatures under dry nitrogen flow until the weight remained stable. The as-synthesized SIFSIX-1-Cu sample was firstly wrapped in aluminum crucibles and directly activated on the PE TGA at 343 K under dry nitrogen flow until the weight remained stable.

The activated samples wrapped in aluminum crucibles were immediately transferred to the sample chamber on the PE DSC 7 to measure the adsorption enthalpy, the baseline was obtained under dry nitrogen flow at 298 K, then the nitrogen flow was changed to hexane isomers bubbled with nitrogen at a certain flow rate, and the DSC signal were monitored to obtain the heat of adsorption at 298 K. After the signal remained stable, the samples wrapped in aluminum crucibles were immediately transferred on the PE TGA to record the adsorbed amount of hexane isomers.

## Computational methods

All calculations were performed by the Materials Studio, 2017R2 package. The first-principle density functional theory (DFT) and plane-wave ultrasoft pseudopotential were implemented in the Materials Studio, CASTEP code[44]. A semi-empirical addition of dispersive forces to conventional DFT was included in the calculation to account for van der Waals interactions. Calculations were performed under the generalized gradient approximation (GGA) with Perdew–Burke–Ernzerhof (PBE) exchange correlation. A cutoff energy of 544 eV and a $2 \times 2 \times 3$ k-point mesh with smearing 0.2 ev were found to be enough for the total energy to converge within $1 \times 10^{-6}$ ev atom$^{-1}$, the calculation error were within 0.15 Å.

**MC simulations under the NVT system.** The preferred adsorption sites of guest molecules were searched through MC simulations under the NVT system in the sorption module. The frameworks of nHEX-loaded ZU-72 framework, zeolite 5A and nHEX molecule would be firstly optimized by DFT-D calculations, and considered to be rigid during the simulation. The charges for atoms of the ZU-72, zeolite 5A, and nHEX were derived from Qeq method and Qeq_charged 1.1 parameters. The simulations adopted the locate task, Metropolis method in sorption module and the universal force field (UFF) for ZU-72, the configurational bias method in the sorption module, and the COMPASSII for zeolite 5A. The interaction energy between the adsorbed molecules and the framework were computed through the Coulomb and Lennard–Jones 6–12 (LJ) potentials. The cutoff radius was chosen 18.5 Å for LJ potential and the long range electrostatic interactions were handled using the Ewald summation method. The loading steps and the equilibration steps were $1 \times 10^7$, the production steps were $1 \times 10^7$.

**The static binding energy.** The frameworks of bare ZU-72 and zeolite 5A would be first optimized by DFT-D calculations. An isolated nHEX molecule placed in a supercell (with the same cell dimensions as the host crystal) was also optimized. The initial host–guest configurations produced by MC simulations under the NVT system was subjected to DFT-D to optimize the host–guest structures, in which the host structure was fixed. The static binding energy (at $T = 0$ K) was then calculated: EB = E (MOF) + E (gas) − E (MOF+ gas). where E (MOF) is the energy of the optimized guest-free host, E (gas) is the energy of the optimized guest and E (MOF+ gas) is the total energy of the optimized host–guest structures.

**The energy landscape of the ZU-72 framework with rotational blocks.** The variation of the total energy as the pyridine ring and $SnF_6^{2-}$ anions rotated around the lattice axis $a/b$ and $c$, respectively, were derived from DFT-D calculations. First, the bare ZU-72 framework was optimized prior to the calculation to set as the reference, and the optimized structure was close to the experimental one determined from SCXRD data. During the calculation, the rotational angle of pyridine rings or $SnF_6^{2-}$ anion was the only variable, while the other atoms were artificially fixed. Then, the single- point energy calculations were performed to calculate the energy of these different rotational angle configurations.

## Data availability

All data supporting the findings of this study are available within this article and its Supplementary Information. Crystallographic data for the structures in this article have been deposited at the Cambridge Crystallographic Data Centre under deposition Nos. CCDC 2279712 (solvent-loaded ZU-72), 2279714 (activated ZU-72), 2279715 (nHEX-loaded ZU-72) and 2279716 (re-activated ZU-72). Copies of the data can be obtained free of charge from www.ccdc.cam.ac.uk/data_request/cif. Source data that support the findings of this study are available from the corresponding author upon request.

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

## Acknowledgements

This work is supported by the National Natural Science Foundation of China (No. 21938011 to H.X., 22122811 to X.C.), the Shanxi-Zheda Institute of Advanced Materials and Chemical Engineering (Grant No. 2021SZ-TD0080 to X.C.), and the National Supercomputing in Shenzhen, Research Computing Center in College of Chemical and Biological Engineering at Zhejiang University.

## Author contributions

H.X., X.C., and Q.W. conceived and designed the research. Q.W. and L.Y. synthesized the materials. Q.W. and T.K. carried out the characterizations and adsorption experiments. Q.W. and X.S. carried out the transient breakthrough measurements. Q.W. and T.K. performed crystal structure determination. Q.W., L.Y., and J.H. performed computational simulations and analyses. Q.W., X.C., and H.X. wrote the manuscript.

## Competing interests

The authors declare no competing interests.
