## [Peer Review File · Nature Communications]

Selective sorting of hexane isomers by anion-functionalized metal-organic frameworks with optimal energy regulationREVIEWER COMMENTS

Reviewer #1 (Remarks to the Author):

This manuscript reports three SIFSIX-based MOFs showing the good performance for separation of alkane isomers. The investigated MOF SNFSIX-2-Cu-i outperform the commercialized zeolite 5A and certain reported MOF materials for the separation of mono- and dibranched alkane isomers. There are studies for the discrimination of branched alkanes from linear alkanes that have been reported, e.g. those by Jing Li's group. Interestingly, two reported MOFs (SIFSIX-1-Cu and ZU-61) were found to separate the challenging 22DMB and 23DMB, in which SIFSIX-1-Cu showed high capacity and excellent di-branched isomers selectivity, further improving the research octane numbers (RONs) of gasoline. The authors have carried out comprehensive experimental characterization for evaluating separation performances of those MOF materials. There is now one major concern on the collected amount of alkanes with qualified RONs (e.g. >92 or 95, during one breakthrough option). And how about this data in comparing with other adsorbent materials?

There are also some issues needed be addressed:

- (1) When "the highest quality gasoline with RON of 105" was claimed, the authors should compare the collected amounts of gasoline, rather than uptake ratios.
- (2) In order for the readership to better understand RON, please indicate in the supporting information how to calculate RON from the eluted mixture in Breakthrough experiments.
- (3) The authors explain that SIFSIX-1-Cu has a size exclusion effect, but do not provide a suitable explanation on size comparison, i.e. giving information about the pore size, etc.
- (4) There are a number of flexible MOFs reported to have inherent thermal management, which the authors should discuss in the introduction section and compare in result section.
- (5) SNFSIX-2-Cu-i and SIFSIX-2-Cu-i are isomers with different metal center. Please compare the separation performances of SIFSIX-2-Cu-i in alkane isomers.
- (6) The yield of these MOFs should be added in the SI.

Reviewer #2 (Remarks to the Author):

In this work, the authors showcase remarkable adsorption and selectivity for various hexane isomers using two different MOF systems, each utilized for its own purpose for separating out different isomers. The results itself speak for itself and the manuscript can be published in the paper with minor revisions.

- 1) In Figure 2(a), the pure component isotherms of various hexane isomers show hysteresis behavior. What is responsible for this (e.g. flexibility, enhanced gas-gas interactions)?
- 2) The authors used GCMC simulations to identify the appropriate binding configurations nHEX in ZU-72.

I presume this was used to get the configurations to use as a starting point for the DFT simulations. But if that is the case, I am curious why GCMC was adopted as a choice. Most common choice would be to use NVT simulations at low temperature to get the ground-state configurations. Did the authors obtain the adsorption isotherms using GCMC simulations and if so, is that relevant to the studies here?

3) I am surprised at the high selectivity of 23DMB vs 22DMB in SIFSIX-1-Cu. Did the authors try computing the DFT binding energies for 23DMB and 22DMB in this MOF to see if there is significant difference in the binding energy that might explain this high selectivity? Or is the high selectivity just due to size separation? The reason why I ask is that the authors spent a lot of effort in elucidating the mechanisms of binding in ZU-72 but not in SIFSIX-1-Cu.

4) In realistic mixture gas, there will be other components (e.g. N₂, CO₂). How would the presence of other gas molecules impact the performance of the system?

Reviewer #3 (Remarks to the Author):

In the submitted manuscript, the authors present a novel approach to tackle the challenge of energy-efficient hexane isomer separation. They introduce a metal-organic framework (MOF) denoted as ZU-72, which exhibits a unique capability to alter its structure during the ad/desorption process due to the flexibility of its organic linker. The authors claimed that this structural transformation results in energy storage and release, leading to enhanced energy efficiency during the separation process. However, there are several critical issues in the manuscript. Thus, I cannot recommend its publication in Nature Communications, a high-profile multidisciplinary journal.

1. Clarity of the Main Idea: The manuscript lacks a clear and cohesive presentation of the central concept. The intended approach involves combining an energy-efficient MOF containing anion-functionalities (ZU-72) in the front, along with well-studied high performing sieves (SIFSIX-1-Cu or ZU-61) in the rear. This strategic combination aims to facilitate hexane isomer separation while enhancing energy efficiency. However, the key fact is that the isomer separation was conducted with SIFSIX-1, which has been widely studied. It is crucial to revise the introduction and title to accurately convey the focus of the research and align them with the actual content of the manuscript. This will prevent potential confusion among readers, although the impact of the new framework is diluted inevitably.

2. Introduction Clarity: A significant improvement is needed in explaining the characteristics of the MOFs discussed in the paper—ZU-72, SIFSIX-1-Cu, and ZU-61. Providing clear differentiations among these MOFs will help readers comprehend the subsequent research content. To enhance the manuscript's readability and coherence, make sure to establish a solid foundation by outlining the unique features of each MOF in the introduction.

3. Novelty and Comparative Analysis: To highlight the novelty of ZU-72, which adds to the anion-functionalized MOF (SIFSIX series) literature, the authors must emphasize its distinctive attributes.

Comparative analyses should be carried out to assess ZU-72's energy consumption for regeneration and its adsorption behavior, in relation to SIFSIX-1-Cu. Moreover, explore how varying anion species influence adsorption performance to provide a comprehensive understanding of the MOF's capabilities. Citations to relevant studies, such as those mentioned below, will support these comparisons.

[1] L. Yang et al., *Chem. Eur. J.*, 2021, 27, 6187-6190 (DOI: 10.1002/chem.202100008)

[2] K. A. Forrest et al., *Cryst. Growth Des.*, 2019, 19, 3732-3743 (DOI: 10.1021/acs.cgd.9b00086)

[3] L. I. Zheqi et al., *Russ. J. Appl. Chem.*, 2014, 87, 1511-1516 (DOI: 10.1134/S1070427214100188)

4. Data Presentation and Figures: Figure 2c requires updating to include an adequate number of comparative groups, considering recent developments in the field. As in Figure 2d, other classes of porous materials can be incorporated for an effective comparison. Also, provide an explanation for the choice of axes in Figure 2c to justify using n-hexane uptake instead of MP uptake.

5. Concluding Remarks: Revise the conclusion to conform to the typical structure of summarizing key results, identifying unresolved issues, and presenting future perspectives. Any new information introduced should be incorporated into the results section, preserving the established structure for the conclusion.

6. Minor Corrections:

- Define abbreviations like RON (Research Octane Number) when they are first mentioned.
- Ensure consistent temperature units (either K or °C) throughout the manuscript and accompanying figures.
- Address the identical nature of Figure 2b and Figure S13.
- In lines 108-111, include references to support the claims made.

Response to reviewers' comments

Comments from Reviewer 1:

This manuscript reports three SIFSIX-based MOFs showing the good performance for separation of alkane isomers. The investigated MOF SNFSIX-2-Cu-i outperform the commercialized zeolite 5A and certain reported MOF materials for the separation of mono- and dibranched alkane isomers. There are studies for the discrimination of branched alkanes from linear alkanes that have been reported, e.g. those by Jing Li's group. Interestingly, two reported MOFs (SIFSIX-1-Cu and ZU-61) were found to separate the challenging 22DMB and 23DMB, in which SIFSIX-1-Cu showed high capacity and excellent di-branched isomers selectivity, further improving the research octane numbers (RONs) of gasoline. The authors have carried out comprehensive experimental characterization for evaluating separation performances of those MOF materials. There is now one major concern on the collected amount of alkanes with qualified RONs (e.g. >92 or 95, during one breakthrough option). And how about this data in comparing with other adsorbent materials?

There are also some issues needed be addressed:

Response: We thank this reviewer for these positive and constructive comments which by addressing have allowed us to improve the manuscript. As suggested, we have calculated the collected amount of alkanes with qualified RONs (>95) on SNFSIX-2-Cu-i (ZU-72) to be 15.8 mL g⁻¹. And for comparison with other adsorbents, we have screened nearly all recent achievements in the separation of five-component hexane isomers and listed the separation performance in the revised Table S1. However, NU-2200, NU-2000, NU-2004 were not tested for the actual experimental breakthrough separations, while Al-bttotb and UU-200 did not provide the complete breakthrough separation conditions, so we were unable to calculate the exact collected amount of alkanes with qualified RONs. Therefore, in this work, we just compared the separation selectivity of SNFSIX-2-Cu-i with these materials, and SNFSIX-2-Cu-i exhibited benchmark selectivity (Figure 2c). Fortunately, Fe₂(BDP)₃ provides the separation conditions, we compared SNFSIX-2-Cu-i with Fe₂(BDP)₃ and SIFSIX-2-Cu-i, ZU-62 studied in this work. As shown in following Figure S32, SNFSIX-2-Cu-i exhibits the highest productivity of high-quality gasoline (RON>95).

Modification: 1) Main text, page 7, line 164-165, added text "The productivity of high-quality gasoline (RON > 95) for ZU-72 (15.8 mL g⁻¹) was higher than that for SIFSIX-2-Cu-i (14.3 mL g⁻¹) and Fe₂(BDP)₃ (1.5 mL g⁻¹) (Figure S32)."

2) Figure S32 has been added to the revised Supplementary Information. Table S1 has been updated in the revised Supplementary Information

Supplementary Figure 32. Comparison of the high quality gasoline (RON>95) productivity during the column breakthrough for the equimolar mixture of nHEX/2MP/3MP/23DMB/22DMB on ZU-72 with other good performing materials.

Table S1. Comparison of the uptake and selectivity of hexane isomers on various materials for five-component isomers separation

	Temperature (K)	Pressure (kPa)	nHEX uptake (mmol g ⁻¹)	2MP uptake (mmol g ⁻¹)	3MP uptake (mmol g ⁻¹)	23DMB uptake (mmol g ⁻¹)	22DMB uptake (mmol g ⁻¹)	Uptake ratio of MP/DMB	Ref.
NU-2004	298	12	1.35	0.79	0.82	0.27	0.26	3.04	[4]
NU-2000	298	12	1.53	0.95	0.87	1.09	0.61	1.07	[4]
NU-2200	298	12	1.28	1.00	0.87	0.19	0.15	5.50	[5]
UU-200	303	12	1.67	1.23	1.12	0.37	0.1	5.00	[6]
Fe ₂ (BPD) ₃	433	16	-	1.12	1.32	1.1	1.09	1.11	[7]
Fe ₂ (BPD) ₃	433	12	1.28	1.04	1.16	0.92	0.92	1.20	[7]
Al-bttotb	303	16	1.74	1.15	1.06	0.42	0.06	4.60	[8]
Al-bttotb	303	12	1.71	1.13	1.04	0.41	0.05	4.72	[8]
ZU-62	298	12	1.71	0.87	0.62	0.15	0.11	5.73	This work
SIFSIX-2-Cu-i	298	12	1.98	1.11	1.05	0.25	0.14	5.53	This work
ZU-72	298	16	2.12	1.19	1.23	0.26	0.15	5.90	This work
ZU-72	298	12	2	1.13	1.17	0.22	0.11	6.97	This work

*MP is defined as (2MP + 3MP)/2;

*DMB is defined as (23DMB + 22DMB)/2

1. When “the highest quality gasoline with RON of 105” was claimed, the authors should compare the collected amounts of gasoline, rather than uptake ratios.

Response: We appreciate the constructive comments from the reviewer. The collected amounts of gasoline in this work, also known as the productivity of an adsorbent, is an important criterion for evaluating the performance of the adsorbent and should be carefully estimated. As suggested, we have calculated the collected amounts of highest quality gasoline with RON of 105 using SIFSIX-1-Cu to be 55.6 mL g⁻¹. However, currently there is no reported exploration on the separation of challenging 22DMB and 23DMB to obtain the highest quality gasoline. Thus, we have not compared this performance factor with other adsorbents.

Modification: Main text, page 7, line 170-172, added text “The following desorption operations enabled the further upgrading of gasoline with the highest RON (~105) (Figure 3c and Figure S35), with a productivity of 55.6 mL g⁻¹”

2. In order for the readership to better understand RON, please indicate in the supporting information how to calculate RON from the eluted mixture in Breakthrough experiments.

Response: As suggested, the detailed calculation of the RON from the eluted mixture in breakthrough experiments has been added in the revised Supplementary Information.

Modification: Supplementary Information, page 4, line 123-137, added the text:

“The calculation of research octane numbers RONs from the eluted mixture in Breakthrough experiments

Firstly, the prepared mixed gases of nHEX/2MP/3MP/23DMB/22DMB (1/1/1/1/1) was monitored by the gas chromatography, and the peak area of each component was recorded as S₀ⁿ, where n=1, 2, 3,4,5 represents nHEX, 2MP, 3MP, 23DMB and 22DMB respectively. Then the outlet gas from the column was monitored by the gas chromatography and the peak area of each component was recorded as Sⁿ, where n=1, 2, 3,4,5 represents for nHEX, 2MP, 3MP, 23DMB and 22DMB respectively. Therefore, the concentration of each component (Cⁿ) can be calculated using the following equation (1):

$$C^n = \frac{S^n/S_0^n}{S^1/S_0^1 + S^2/S_0^2 + S^3/S_0^3 + S^4/S_0^4 + S^5/S_0^5} \quad (1)$$

The RON of nHEX, 2MP, 3MP, 23DMB and 22DMB are 30, 74, 75, 94 and 105 respectively, therefore the RON of the outlet mixture gas can be calculated using the following equation (2):

$$RON = 30 \times C^1 + 74 \times C^2 + 75 \times C^3 + 94 \times C^4 + 105 \times C^5 \quad (2)$$

3. The authors explain that SIFSIX-1-Cu has a size exclusion effect, but do not provide a suitable explanation on size comparison, i.e. giving information about the pore size, etc.

Response: We appreciate the constructive comments from the reviewer. The pore information and crystal structure of SIFSIX-1-Cu have been added to the revised manuscript and Supplementary Information. The molecular sizes of 22DMB and 23DMB are 8.0 x 6.7 x 5.9 Å³ and 7.8 x 6.7 x 5.3 Å³, respectively. As shown in the following Figure S2, the pore size of SIFSIX-1-Cu falls within

the range of 6.8 Å - 7.2 Å, potentially resulting in a size exclusion effect for the relative larger molecule of 22DMB (8.0 x 6.7 x 5.9 Å³). Additionally, the crystal size of SIFSIX-1-Cu is in the micrometer range, with dimensions reaching up to 15 μm (Figure S3). Such large crystal sizes would hinder the diffusion of molecules with sizes similar to the pores and enhance the size exclusion effect. Therefore, as demonstrated by the single-component equilibrium isotherm (Figure 2b), differential scanning calorimetry experiments (Figure S18), dynamic adsorption tests (Figure S19), and actual breakthrough tests (Figure 3b), 22DMB cannot enter the pore of SIFSIX-1-Cu, resulting in the molecular sieving effect for 22DMB. The corresponding discussion have been added in the revised manuscript.

Modification: 1) Main text, page 4, line 102-106, added text “The non-interpenetrated SIFSIX-1-Cu (1 = 4,4'-dipyridine) is constructed by six-copper node coordinated with four organic linkers and then pillared with SiF₆²⁻ anions, with pore sizes distributed around 6.8 Å - 7.2 Å (Figure S2), larger than that of ZU-72. ZU-61 (also termed NbOFFIVE-1-Ni) is isomorphic to SIFSIX-1-Cu, with slightly wider pore sizes distributed around 7.3 Å - 7.8 Å (Figure S4)”.

2) Main text, page 5, line 125-127, updated text “SIFSIX-1-Cu with suitable pore size and large crystal size realizes the size exclusion effect of 22DMB (8.0 x 6.7 x 5.9 Å³) from 23DMB (7.8 x 6.7 x 5.3 Å³)”.

3) Figure S2 and S3 have been added in the revised Supplementary Information.

Supplementary Figure 2. Crystal Structures of SIFSIX-1-Cu. Color code: F, red; Si, light orange; Cu, light blue; C, gray- 40%; H, gray-25%; N, blue.

Supplementary Figure 3. Crystal morphology of SIFSIX-1-Cu.

4. There are a number of flexible MOFs reported to have inherent thermal management, which the authors should discuss in the introduction section and compare in result section.

Response: We thank this reviewer for the important comments. As suggested, we have included some flexible MOFs reported to have inherent thermal management in the introduction section and compare in result section.

Modification: 1) Main text, page 3, line 58-62, added the text: “Recently, there are a number of flexible MOFs reported to have inherent thermal management applied to achieve highly efficient separation/storage gases²³⁻²⁹. These materials can reversibly utilize latent heat during the phase change process for thermal energy storage and release, enabling efficient adsorption-desorption cycles.”

2) Main text, page 9, line 218-219, added the text: “The intrinsic thermal management capability of ZU-72 is relative higher than the Fe(bdp) (8.1 kJ mol⁻¹) and Co(bdp) (7.0 kJ mol⁻¹) utilized for CH₄ storage²³.”

3) Some relevant citations have been updated in the references as below:

“23. Mason, J. A. et al. Methane storage in flexible metal–organic frameworks with intrinsic thermal management. *Nature* **527**, 357-361 (2015).

24. Coudert, F. X., Jeffroy M., Fuchs, A. H., Boutin, A. & Mellot-Draznieks, C. Thermodynamics of Guest-Induced Structural Transitions in Hybrid Organic-Inorganic Frameworks. *J. Am. Chem. Soc.* **130**, 14294-14302 (2008).

25. Hiraide, S., Tanaka, H., Ishikawa, N. & Miyahara, M. T. Intrinsic thermal management capabilities of flexible metal–organic frameworks for carbon dioxide separation and capture. *ACS Appl. Mater. Interfaces* **9**, 41066–41077 (2017).

26. Snyder, B. E. R. et al. A ligand insertion mechanism for cooperative NH₃ capture in metal-organic frameworks. *Nature* **613**, 287-291 (2023).

27. Hiraide, S. et al. High-throughput gas separation by flexible metal–organic frameworks with fast gating and thermal management capabilities. *Nat Commun.* **11**, 3867 (2020).

28. Liao, P. Q. et al. Controlling guest conformation for efficient purification of butadiene. *Science* **356**, 1193-1196 (2017).

29. Horike, S., Shimomura, S. & Kitagawa, S. Soft porous crystals. *Nat. Chem.* **1**, 695-704 (2009).”

5. SNFSIX-2-Cu-i and SIFSIX-2-Cu-i are isomers with different metal center. Please compare the separation performances of SIFSIX-2-Cu-i in alkane isomers.

Response: As suggested, in order to compare the separation performances of SIFSIX-2-Cu-i in alkane isomers, single-component adsorption isotherms and experimental breakthrough curves have been conducted on SIFSIX-2-Cu-i and the corresponding discussion have been added in the revised manuscript. As shown in the following figures, SIFSIX-2-Cu-i and SNFSIX-2-Cu-i are isostructural, but with slightly different pore sizes, 4.70 Å for SIFSIX-2-Cu-i, and 4.59 Å for ZU-72. Both materials can discriminate the hexane isomers according to the degree of branching. However, SNFSIX-2-Cu-i exhibits the higher separation selectivity (6.97) compared with SIFSIX-2-Cu-i (5.53) (Figure 2c and S21). And experimental breakthrough curves indicate SNFSIX-2-Cu-i has a higher productivity of high-quality gasoline (RON > 95) (15.8 mL g⁻¹) compared to SIFSIX-2-Cu-i (14.3 mL g⁻¹) (Figure S32).

Figure. The crystal structures of S SIFSIX-2-Cu-i (a) and SNFSIX-2-Cu-i (b), respectively. Pure-component vapor adsorption isotherms of n-hexane (nHEX), 2-methylpentane (2MP), 3-methylpentane (3MP), 2,3-dimethylbutane (23DMB) and 2,2-dimethylpentane (22DMB) on SIFSIX-2-Cu-i (c) and SNFSIX-2-Cu-i (d), respectively at 298 K. Experimental breakthrough curves for the equimolar mixture of nHEX/2MP/3MP/23DMB/22DMB on SIFSIX-2-Cu-i (e), and SNFSIX-2-Cu-i (d), respectively at 298 K.

Modification: 1) Main text, page 5, line 117-120, added text “endowing ZU-72 with the record separation selectivity of mono-/di-branched (6.97), higher than the isostructural SIFSIX-2-Cu-i (5.53) and previous benchmark materials, $\text{Fe}_2(\text{BDP})_3$ (1.21)⁵ and Al-bttotb (4.72)²¹ (Figure 2c and Figure S21).”

2) Main text, page 7, line 164-165, added text “The productivity of high-quality gasoline (RON > 95) for ZU-72 (15.8 mL g⁻¹) was higher than that for SIFSIX-2-Cu-i (14.3 mL g⁻¹) and $\text{Fe}_2(\text{BDP})_3$ (1.5 mL g⁻¹) (Figure S32).”

3) Figure 2 has been updated in the revised manuscript. Figure S38, Figure 32 and Figure S21 have been added in the revised Supplementary Information.

Supplementary Figure 38. The comparison of crystal structures and hexane isomers separation performance of varying anion functionalized MOFs. The crystal structures of SiF_6^{2-} , SnF_6^{2-} and NbOF_6^{2-} pillared MOFs, namely SIFSIX-2-Cu-i (a), ZU-72 (b) and ZU-62 (c), respectively. Pure-component vapor adsorption isotherms of n-hexane (nHEX), 2-methylpentane (2MP), 3-methylpentane (3MP), 2,3-dimethylbutane (23DMB) and 2,2-dimethylpentane (22DMB) on SIFSIX-2-Cu-i (d), ZU-72 (e) and ZU-62 (f), respectively at 298 K. Experimental breakthrough curves for the equimolar mixture of nHEX/2MP/3MP/23DMB/22DMB on SIFSIX-2-Cu-i (g), ZU-72 (h) and ZU-62 (i), respectively at 298 K. Experimental breakthrough curves for the equimolar mixture of nHEX/2MP/3MP/23DMB/22DMB on ZU-72 together with the RON calculated from the eluted mixture SIFSIX-2-Cu-i (j), ZU-72 (k) and ZU-62 (l), respectively at 298 K.

Fig. 2. Hexane isomers adsorption isotherms of anion-functionalized MOFs. Pure-component vapor adsorption isotherms of nHEX, 2MP, 3MP, 23DMB and 22DMB on ZU-72 at 298 K in the region of 0-16 kPa (a). Pure-component vapor adsorption isotherms of 23DMB and 22DMB on SIFSIX-1-Cu at 298 K in the region of 0-16 kPa (b). Plots of the nHEX uptake as a function of the MP/DMB uptake ratio for ZU-72 and other previously reported materials at 298 K and 12 kPa (c). (d) Plots of the 23DMB uptake at 12 kPa as a function of the 23DMB/22DMB uptake ratio for SIFSIX-1-Cu and other previously reported materials at 298 K. ^a at 303 K, ^b at 423 K and ^c at 433 K. MP/DMB is defined as (2MP + 3MP)/(23DMB+22DMB).

Supplementary Figure 21. Plots of the MP uptake as a function of the MP/DMB uptake ratio for ZU-72 and other previously reported materials at 298 K and 12 kPa. ^a at 303 K, ^b at 433 K. MP/DMB is defined as (2MP + 3MP)/(23DMB+22DMB).

Supplementary Figure 32. Comparison of the high quality gasoline (RON>95) productivity during the column breakthrough for the equimolar mixture of nHEX/2MP/3MP/23DMB/22DMB on ZU-72 with other good performing materials.

6. *The yield of these MOFs should be added in the SI.*

Response: The yield of ZU-72, SIFSIX-1-Cu and ZU-61 are 70 %, 77% and 65 %, respectively. These data have been added in the revised Supplementary Information.

Modification: The text “The yield of ZU-72 is about 70%”, “The yield of SIFSIX-1-Cu is about 77%” and “The yield of ZU-61 is about 65%” have been added in the revised Supplementary Information.

Comments from Reviewer 2:

In this work, the authors showcase remarkable adsorption and selectivity for various hexane isomers using two different MOF systems, each utilized for its own purpose for separating out different isomers. The results itself speak for itself and the manuscript can be published in the paper with minor revisions.

Response: We thank this referee for these positive comments.

1. *In Figure 2(a), the pure component isotherms of various hexane isomers show hysteresis behavior. What is responsible for this (e.g. flexibility, enhanced gas-gas interactions)?*

Response: In this work, the hysteresis behavior observed in the adsorption isotherms of hexane isomers on ZU-72 is attributed to the flexibility of the structure. As revealed by single-crystal X-ray diffraction analysis and simulations, the pyridine rings and SnF₆²⁻ anions undergo flexible rotation upon adsorption of hexane isomers. It is common for flexible materials to exhibit hysteresis phenomena in the adsorption procedure. And the corresponding discussion has been added in the revised manuscript.

Modification: 1) Main text, page 8, line 199-200, added text “The flexibility of the structure accounts for the hysteresis behavior observed in the adsorption isotherms.”

2. The authors used GCMC simulations to identify the appropriate binding configurations nHEX in ZU-72. I presume this was used to get the configurations to use as a starting point for the DFT simulations. But if that is the case, I am curious why GCMC was adopted as a choice. Most common choice would be to use NVT simulations at low temperature to get the ground-state configurations. Did the authors obtain the adsorption isotherms using GCMC simulations and if so, is that relevant to the studies here?

Response: i) Many thanks for the reviewer pointing out the incorrect use "GCMC simulations", we may have made a clerical mistake here. Actually, the simulations we used to obtain the initial configurations were based on NVT system, i.e., *Locate/Fixed loading* tasks in the *Sorption* module of the *Material Studio* software. These tasks were performed under the defined particle number, volume, and temperature. We have revised the related description "GCMC simulations" to "MC simulations under the NVT system".

The index of the task from *Material Studio* used for MC simulations was provided:

Locate task: The *Locate* task in *Sorption* searches a sorbent framework to find the preferential, that is, lowest energy, sites for sorbate inclusion. A Monte Carlo search of the configurational space of the sorbate-sorbent system is carried out while the temperature is slowly decreased. This process is repeated to ensure comprehensive coverage of the energy surface.

ii) The adsorption isotherms presented in the manuscript and SI were experimental data, and we had not performed computational simulations of GCMC to obtain the adsorption isotherms.

Modification: The "GCMC simulations" have been corrected to "MC simulations under the NVT system" in the revised manuscript and Supplementary Information.

3. I am surprised at the high selectivity of 23DMB vs 22DMB in SIFSIX-1-Cu. Did the authors try computing the DFT binding energies for 23DMB and 22DMB in this MOF to see if there is significant difference in the binding energy that might explain this high selectivity? Or is the high selectivity just due to size separation? The reason why I ask is that the authors spent a lot of effort in elucidating the mechanisms of binding in ZU-72 but not in SIFSIX-1-Cu.

Response: We thank the reviewer for these important comments. The high selectivity of 23DMB vs 22DMB in SIFSIX-1-Cu is mainly attributed to the size exclusion effect. We didn't perform the DFT calculation to evaluate the binding energies, but instead used experimental methods such as differential scanning calorimetry test to assess it, where no adsorption heat of 23DMB is detected, in comparison with 63 kJ/mol for 22DMB (Figure S18), thereby thermodynamically demonstrating the size exclusion effect of SIFSIX-1-Cu towards 22DMB. The single-component equilibrium isotherm (Figure 2b), dynamic adsorption tests (Figure S19), and actual breakthrough tests (Figure 3b) also demonstrate that 22DMB cannot enter the pore of SIFSIX-1-Cu. As shown in the following Figure S2, the pore size of SIFSIX-1-Cu falls within the range of 6.8 Å - 7.2 Å, potentially resulting in a size exclusion effect for the relative larger molecule of 22DMB (8.0 x 6.7 x 5.9 Å³). The crystal size of SIFSIX-1-Cu is in the micrometer range, with dimensions reaching up to 15 μm (Figure S3). Such large crystal sizes would hinder the diffusion of molecules with sizes similar to the pores and enhance the size exclusion effect, resulting in the high selectivity of 23DMB vs 22DMB in SIFSIX-1-Cu. The corresponding discussion have been added in the revised manuscript.

Modification: 1) Main text, page 4, line 102-106, added text “The non-interpenetrated SIFSIX-1-Cu (1 = 4,4'-dipyridine) is constructed by six-copper node coordinated with four organic linkers and then pillared with SiF_6^{2-} anions, with pore sizes distributed around 6.8 Å - 7.2 Å (Figure S2), larger than that of ZU-72. ZU-61 (also termed NbOFFIVE-1-Ni) is isomorphic to SIFSIX-1-Cu, with slightly wider pore sizes distributed around 7.3 Å - 7.8 Å (Figure S4)”.

2) Main text, page 5, line 125-127, updated text “SIFSIX-1-Cu with suitable pore size and large crystal size realizes the molecular exclusion of 22DMB ($8.0 \times 6.7 \times 5.9 \text{ \AA}^3$) from 23DMB ($7.8 \times 6.7 \times 5.3 \text{ \AA}^3$)”.

3) Figure S2 and S3 have been added in the revised Supplementary Information.

Supplementary Figure 2. Crystal Structures of SIFSIX-1-Cu. Color code: F, red; Si, light orange; Cu, light blue; C, gray- 40%; H, gray-25%; N, blue.

Supplementary Figure 3. Crystal morphology of SIFSIX-1-Cu.

4. *In realistic mixture gas, there will be other components (e.g. N_2 , CO_2). How would the presence of other gas molecules impact the performance of the system?*

Response: We thank the reviewer for these constructive comments. As suggested, in order to investigate the influence of the presence of other gas components including N_2 , O_2 and CO_2 , single-component adsorption isotherms of N_2 , O_2 and CO_2 on ZU-72, SIFSIX-1-Cu and ZU-61 are conducted and included in the revised manuscript. As shown in the following Figure S25, ZU-72, SIFSIX-1-Cu and ZU-61 exhibit negligible uptakes towards N_2 and O_2 , and adsorb negligible uptake towards CO_2 at low concentration of 400 ppm (mimicking air concentration) at 298 K, demonstrating high selectivities towards hexane isomers over these impurities (Supplementary Figure 25).

Therefore, these gas impurities have no noticeable effect on these materials for the separation of hexane isomers.

Modification: Supplementary Figure 25 has been added in the revised Supplementary Information.

Supplementary Figure 25. The adsorption isotherms of N₂, O₂, CO₂ and hexane isomers on ZU-72, SIFSIX-1-Cu and ZU-61 at 298 K.

Comments from Reviewer 3:

In the submitted manuscript, the authors present a novel approach to tackle the challenge of energy-efficient hexane isomer separation. They introduce a metal-organic framework (MOF) denoted as ZU-72, which exhibits a unique capability to alter its structure during the ad/desorption process due to the flexibility of its organic linker. The authors claimed that this structural transformation results in energy storage and release, leading to enhanced energy efficiency during the separation process. However, there are several critical issues in the manuscript. Thus, I cannot recommend its publication in Nature Communications, a high-profile multidisciplinary journal.

Response: We thank the reviewer for these constructive comments and recognizing the novelty of our work, which by addressing have allowed us to improve the manuscript.

1. Clarity of the Main Idea: The manuscript lacks a clear and cohesive presentation of the central concept. The intended approach involves combining an energy-efficient MOF containing anion-functionalities (ZU-72) in the front, along with well-studied high performing sieves (SIFSIX-1-Cu or ZU-61) in the rear. This strategic combination aims to facilitate hexane isomer separation while enhancing energy efficiency. However, the key fact is that the isomer separation was conducted with

SIFSIX-1, which has been widely studied. It is crucial to revise the introduction and title to accurately convey the focus of the research and align them with the actual content of the manuscript. This will prevent potential confusion among readers, although the impact of the new framework is diluted inevitably.

Response: We thank the reviewer for the important and constructive comments. In this work, SIFSIX-1-Cu indeed exhibits excellent separation performance for the di-branched alkanes (22DMB and 23DMB), but its functionality is limited in practical application for five-component hexane isomers, as it cannot achieve the crucial separation of mono- and di-branched alkanes, however, the energy-efficient sorting of which is the key challenge to be pursued. Here, we put forward an approach that rationally controlling selective binding sites and pore size together with flexible pore space within anion-functionalized metal-organic frameworks (ZU-72), to achieve the highly efficient separation of alkane isomers. ZU-72 with suitable pore size and structural flexibility achieves the record mono-/di-branched selectivity and the low regeneration energy. The main innovations of this work are as follows:

(i) First, the record separation selectivity of five-component hexane isomers mixtures on ZU-72. ZU-72 with suitable pore size realizes the molecular size sieve of di-branched isomers with record mono-/di-branched selectivity (6.97).

(ii) Second, the low regeneration energy of ZU-72. The essence of our strategy is to design reversible adaptive frameworks and exert the intrinsic energy regulation in ZU-72 to lower the regeneration energy. ZU-72 can be easily regenerated only through a simple vacuum treatment for 15 minutes at 25 °C with no temperature fluctuation, saving almost 45% energy compared to the commercialized zeolite 5A.

(iii) Third, the first investigation to separate the similar di-branched hexane isomers 23DMB and 22DMB to further upgrade gasoline RON to 105 using SIFSIX-1-Cu.

As suggested, to better convey the central concept of our work, we have revised the title and introduction of the article in the revised manuscript.

Modification: 1) The title has been changed to “Selective sorting of hexane isomers by anion-functionalized metal-organic frameworks with optimal energy regulation”

2) Main text, page 2, line 51-53: added text “Additionally, within these five components, 23DMB has the highest RON (105) and needs further purification to obtain the highest quality gasoline, meeting more application scenarios”.

2. Introduction Clarity: A significant improvement is needed in explaining the characteristics of the MOFs discussed in the paper—ZU-72, SIFSIX-1-Cu, and ZU-61. Providing clear differentiations among these MOFs will help readers comprehend the subsequent research content. To enhance the manuscript's readability and coherence, make sure to establish a solid foundation by outlining the unique features of each MOF in the introduction.

Response: We thank the reviewer for these valuable comments. As suggested, we have added the detailed information about the three anion-functionalized MOFs in the revised manuscript, and added the structure of SIFSIX-1-Cu and ZU-61 in the revised Supplementary Information.

SNFSIX-2-Cu-i (ZU-72), SIFSIX-1-Cu, and ZU-61 (NBOFFIVE-1-Ni) are all anion-pillared ultramicroporous materials, with their three-dimensional frameworks initially formed by the

coordination of metal nodes with organic ligands to form two-dimensional grids, and further supported by anions to form three-dimensional framework structures. Specifically, SNFSIX-2-Cu-i is formed by six-copper node coordinated with four organic linkers (4,4-dipyridylacetylene) and then pillared with SnF_6^{2-} anions. For the long organic linker, the three-dimensional framework will further form an interpenetrated structure to stabilize the framework, where independent nets intertwine with each other to create a dual-interpenetrating structure. However, the organic ligands of SIFSIX-1-Cu and ZU-61 are shorter (4,4'-bipyridine), which do not form interpenetrating structures. The non-interpenetrated SIFSIX-1-Cu (1 = 4,4'-dipyridine) is constructed by six-copper node coordinated with four organic linkers and then pillared with SiF_6^{2-} anions, with pore sizes distributed around 6.8 Å - 7.2 Å, larger than that of ZU-72. ZU-61 (also termed NbOFFIVE-1-Ni) is isomorphic to SIFSIX-1-Cu, with slightly wider pore sizes distributed around 7.3 Å - 7.8 Å. The corresponding discussion have been added in the revised manuscript.

Modification: 1) Main text, page 4, line 93-106: added text “Thus, the interpenetrated anion-functionalized MOFs ZU-72 (ZU = Zhejiang University, also termed SNFSIX-2-Cu-i, 2 = 4,4-dipyridylacetylene, i = interpenetrated), isomorphous to the SIFSIX-2-Cu-i, with pore sizes distributed around 3.6-5.9 Å, are synthesized. The three-dimensional frameworks are formed by six-copper node coordinated with four organic linkers and then pillared with SnF_6^{2-} anions (Figure S1a).” and “The non-interpenetrated SIFSIX-1-Cu (1 = 4,4'-dipyridine) is constructed by six-copper node coordinated with four organic linkers and then pillared with SiF_6^{2-} anions, with pore sizes distributed around 6.8 Å - 7.2 Å (Figure S2), larger than that of ZU-72. ZU-61 (also termed NbOFFIVE-1-Ni) is isomorphic to SIFSIX-1-Cu, with slightly wider pore sizes distributed around 7.3 Å - 7.8 Å (Figure S4)”.

2) Figure S2 and S4 have been added in the revised Supplementary Information.

Supplementary Figure 2. Crystal Structures of SIFSIX-1-Cu. Color code: F, red; Si, light orange; Cu, light blue; C, gray- 40%; H, gray-25%; N, blue.

Supplementary Figure 4. Crystal Structures of ZU-61. Color code: F, red; O, dark red; Nb, cyan; Ni, steel blue; C, gray- 40%; H, gray-25%; N, blue.

3. Novelty and Comparative Analysis: To highlight the novelty of ZU-72, which adds to the anion-functionalized MOF (SIFSIX series) literature, the authors must emphasize its distinctive attributes. Comparative analyses should be carried out to assess ZU-72's energy consumption for regeneration and its adsorption behavior, in relation to SIFSIX-1-Cu. Moreover, explore how varying anion species influence adsorption performance to provide a comprehensive understanding of the MOF's capabilities. Citations to relevant studies, such as those mentioned below, will support these comparisons.

[1] L. Yang et al., *Chem. Eur. J.*, 2021, 27, 6187-6190 (DOI: 10.1002/chem.202100008)

[2] K. A. Forrest et al., *Cryst. Growth Des.*, 2019, 19, 3732-3743 (DOI: 10.1021/acs.cgd.9b00086)

[3] L. I. Zheqi et al., *Russ. J. Appl. Chem.*, 2014, 87, 1511-1516 (DOI: 10.1134/S1070427214100188)

Response: We thank the reviewer for the important comment. As suggested, we have performed comparative analyses and discussed the influence of different anion species on the adsorption performance, which indeed emphasize the distinctive attributes of the SnF_6^{2-} anion and highlight the novelty of ZU-72 (SNFSIX-2-Cu-i). In this work, ZU-72 and SIFSIX-1-Cu have different roles, ZU-72 focuses on addressing the challenges in separating complex five-component mixtures (nHEX/2MP/3MP/23DMB/22DMB), while SIFSIX-1-Cu is utilized for the separation of the di-branched isomers (23DMB/22DMB). Compared to SIFSIX-1-Cu, ZU-72 has a significantly smaller pore size and exhibits distinct flexible adsorption behavior towards hexane isomers. Due to the inherent thermal management properties of flexible materials, ZU-72 has very low regeneration energy consumption and can be regenerated at room temperature. However, SIFSIX-1-Cu with a larger pore size shows a typical Langmuir adsorption isotherm for linear, mono-branched hexane isomers and 23DMB, and exhibits a steeper adsorption curve at low pressures due to the strong host-guest interactions. Therefore, compared to ZU-72, regeneration of SIFSIX-1-Cu is more challenging and requiring 343 K to achieve complete regeneration.

Moreover, to gain a deeper understanding of the impact of different anion species on the adsorption performance, we have explored some other anion-pillared isostructures. As shown in the following Figure S38, the separation performance vary with different anions (SiF_6^{2-} , SnF_6^{2-} and NbO_5^{2-}) owing to the subtle structural differences. Among them, SNFSIX-2-Cu-i exhibits the benchmark separation selectivity and adsorption capacity (Figure 2c and S21), and possess the

highest productivity of high-quality gasoline (RON >95) (15.8 mL g⁻¹) compared to SIFSIX-2-Cu-i (14.3 mL g⁻¹) and NbOFFIVE-2-Cu-i (0.8 mL g⁻¹).

Modification: 1) Main text, page 5, line 123-125, added text “The distinctive attributes of ZU-72 is the introduce of SnF₆²⁻ anion compared with other anion-functionalized isostructures (Figure S39)³³⁻³⁵”

2) Main text, page 5, line 117-120, added text “endowing ZU-72 with the record separation selectivity of mono-/di-branched (6.97), higher than the isostructural SIFSIX-2-Cu-i (5.53) and previous benchmark materials, Fe₂(BDP)₃ (1.21)⁵ and Al-bttotb (4.72)²¹ (Figure 2c and Figure S21).”

3) Main text, page 7, line 164-165, added text “The productivity of high-quality gasoline (RON > 95) for ZU-72 (15.8 mL g⁻¹) was higher than that for SIFSIX-2-Cu-i (14.3 mL g⁻¹) and Fe₂(BDP)₃ (1.5 mL g⁻¹) (Figure S32).”

4) Main text, page 16, line 472-473, added text “Specifically, the temperatures of regeneration are 298 K for ZU-72, 343 K for SIFSIX-1-Cu and 358 K for ZU-61.

5) The relevant references have been cited in the revised manuscript.

6) Figure 2 has been updated in the revised manuscript. Figure S38, Figure S32 and Figure S21 have been added in the revised Supplementary Information.

Supplementary Figure 38. The comparison of crystal structures and hexane isomers separation performance of varying anion functionalized MOFs. The crystal structures of SiF_6^{2-} , SnF_6^{2-} and NbOF_6^{2-} pillared MOFs, namely SIFSIX-2-Cu-i (a), ZU-72 (b) and ZU-62 (c), respectively. Pure-component vapor adsorption isotherms of n-hexane (nHEX), 2-methylpentane (2MP), 3-methylpentane (3MP), 2,3-dimethylbutane (23DMB) and 2,2-dimethylpentane (22DMB) on SIFSIX-2-Cu-i (d), ZU-72 (e) and ZU-62 (f), respectively at 298 K. Experimental breakthrough curves for the equimolar mixture of nHEX/2MP/3MP/23DMB/22DMB on SIFSIX-2-Cu-i (g), ZU-72 (h) and ZU-62 (i), respectively at 298 K. Experimental breakthrough curves for the equimolar mixture of nHEX/2MP/3MP/23DMB/22DMB on ZU-72 together with the RON calculated from the eluted mixture SIFSIX-2-Cu-i (j), ZU-72 (k) and ZU-62 (l), respectively at 298 K.

Fig. 2. Hexane isomers adsorption isotherms of anion-functionalized MOFs. Pure-component vapor adsorption isotherms of nHEX, 2MP, 3MP, 23DMB and 22DMB on ZU-72 at 298 K in the region of 0-16 kPa (a). Pure-component vapor adsorption isotherms of 23DMB and 22DMB on SIFSIX-1-Cu at 298 K in the region of 0-16 kPa (b). Plots of the nHEX uptake as a function of the MP/DMB uptake ratio for ZU-72 and other previously reported materials at 298 K and 12 kPa (c). (d) Plots of the 23DMB uptake at 12 kPa as a function of the 23DMB/22DMB uptake ratio for SIFSIX-1-Cu and other previously reported materials at 298 K. ^a at 303 K, ^b at 423 K and ^c at 433 K. MP/DMB is defined as (2MP + 3MP)/(23DMB+22DMB).

Supplementary Figure 21. Plots of the MP uptake as a function of the MP/DMB uptake ratio for ZU-72 and other previously reported materials at 298 K and 12 kPa. ^a at 303 K, ^b at 433 K. MP/DMB is defined as (2MP + 3MP)/(23DMB+22DMB).

Supplementary Figure 32. Comparison of the high quality gasoline (RON>95) productivity during the column breakthrough for the equimolar mixture of nHEX/2MP/3MP/23DMB/22DMB on ZU-72 with other good performing materials.

4. Data Presentation and Figures: Figure 2c requires updating to include an adequate number of comparative groups, considering recent developments in the field. As in Figure 2d, other classes of porous materials can be incorporated for an effective comparison. Also, provide an explanation for the choice of axes in Figure 2c to justify using n-hexane uptake instead of MP uptake.

Response: We thank the reviewer for the important comment. As suggested, we have screened almost all recent achievements in this field, and listed the separation performance in the revised Table S1 and S2. To provide a more comprehensive comparison, we have revised Figure 2c with recently reported materials aimed to separate five-component mixtures and other anion-functionalized MOFs in this work, and also added Figure S21 with x-axis representing the MP uptake. It can be observed ZU-72 exhibits excellent separation performance with record selectivity (MP/DMB) and high uptake of linear and mono-branched hexane isomers (Figure 2c and S21). We have revised Figure 2d and incorporated almost all the reported data.

Modification: 1) Main text, page 5, line 117-120, added text “endowing ZU-72 with the record separation selectivity of mono-/di-branched (6.97), higher than the isostructural SIFSIX-2-Cu-i (5.53) and previous benchmark materials, Fe₂(BDP)₃ (1.21)⁵ and Al-bttotb (4.72)²¹ (Figure 2c and Figure S21).”

2) Figure 2c, Figure 2d, Table S1 and Table S2 have been updated in the revised manuscript and Supplementary Information. Figure S21 has been added in the revised Supplementary Information

Fig. 2. Hexane isomers adsorption isotherms of anion-functionalized MOFs. Pure-component vapor adsorption isotherms of nHEX, 2MP, 3MP, 23DMB and 22DMB on ZU-72 at 298 K in the region of 0-16 kPa (a). Pure-component vapor adsorption isotherms of 23DMB and 22DMB on SIFSIX-1-Cu at 298 K in the region of 0-16 kPa (b). Plots of the nHEX uptake as a function of the MP/DMB uptake ratio for ZU-72 and other previously reported materials at 298 K and 12 kPa (c). (d) Plots of the 23DMB uptake at 12 kPa as a function of the 23DMB/22DMB uptake ratio for SIFSIX-1-Cu and other previously reported materials at 298 K. ^a at 303 K, ^b at 423 K and ^c at 433 K. MP/DMB is defined as (2MP + 3MP)/(23DMB+22DMB).

Supplementary Figure 21. Plots of the MP uptake as a function of the MP/DMB uptake ratio for ZU-72 and other previously reported materials at 298 K and 12 kPa. ^a at 303 K, ^b at 433 K. MP/DMB is defined as (2MP + 3MP)/(23DMB+22DMB).

Table S1. Comparison of the uptake and selectivity of hexane isomers on various materials for five-component isomers separation

	Temperature (K)	Pressure (kPa)	nHEX uptake (mmol g ⁻¹)	2MP uptake (mmol g ⁻¹)	3MP uptake (mmol g ⁻¹)	23DMB uptake (mmol g ⁻¹)	22DMB uptake (mmol g ⁻¹)	Uptake ratio of MP/DMB	Ref.
NU-2004	298	12	1.35	0.79	0.82	0.27	0.26	3.04	[4]
NU-2000	298	12	1.53	0.95	0.87	1.09	0.61	1.07	[4]
NU-2200	298	12	1.28	1.00	0.87	0.19	0.15	5.50	[5]
UU-200	303	12	1.67	1.23	1.12	0.37	0.1	5.00	[6]
Fe ₂ (BPD) ₃	433	16	-	1.12	1.32	1.1	1.09	1.11	[7]
Fe ₂ (BPD) ₃	433	12	1.28	1.04	1.16	0.92	0.92	1.20	[7]
Al-bttotb	303	16	1.74	1.15	1.06	0.42	0.06	4.60	[8]
Al-bttotb	303	12	1.71	1.13	1.04	0.41	0.05	4.72	[8]
ZU-62	298	12	1.71	0.87	0.62	0.15	0.11	5.73	This work
SIFSIX-2-Cu-i	298	12	1.98	1.11	1.05	0.25	0.14	5.53	This work
ZU-72	298	16	2.12	1.19	1.23	0.26	0.15	5.90	This work
ZU-72	298	12	2	1.13	1.17	0.22	0.11	6.97	This work

*MP is defined as (2MP + 3MP)/2;

*DMB is defined as (23DMB + 22DMB)/2

Table S2. Comparison of the uptake and selectivity of hexane isomers on various materials

	Temperature (K)	Pressure (kPa)	nHEX uptake (mmol g ⁻¹)	2MP uptake (mmol g ⁻¹)	3MP uptake (mmol g ⁻¹)	23DMB uptake (mmol g ⁻¹)	22DMB uptake (mmol g ⁻¹)	Uptake ratio of 23MB/22DMB	Ref.
Ca(H ₂ tcpb)	303	12	1.77	-	1.93	-	1.16	-	[9]
CAU-10-H/Br	303	12	1.57	-	0.75	-	0.12	-	[10]
MOF-1	313	16	0.41	-	0.08	-	0.06	-	[11]
MOF-1	313	12	0.36	-	0.06	-	0.05	-	[11]
Zn ₂ (Hbdc)(dmtrz) ₂	298	16	1.5	-	1.3	-	0.23	-	[12]
Zn ₂ (Hbdc)(dmtrz) ₂	298	12	1.43	-	1.26	-	0.2	-	[12]
MIL-53(Fe)-(CF ₃) ₂	313	16	0.32	-	0.29	-	0.23	-	[13]
MIL-53(Fe)-(CF ₃) ₂	313	12	0.3	-	0.27	-	0.21	-	[13]
HIAM-302	303	16	1.84	-	1.01	-	0.12	-	[14]
HIAM-302	303	12	1.66	-	0.94	-	0.11	-	[14]
HIAM-203	303	16	1.57	-	1.37	-	0.06	-	[15]
HIAM-203	303	12	1.55	-	1.36	-	0.05	-	[15]
Zn-tcpt	303	16	3.09	-	2.05	-	0.05	-	[16]
Zn-tcpt	303	12	2.95	-	2.02	-	0.04	-	[16]
UIO-66	303	12	2.66	-	2.66	2.65	-	-	[17]
Zr-abtc	303	12	2.49	-	2.07	1.1	-	-	[18]
1	298	12	0.86	-	0.63	-	0.46	-	[19]
Ni-Asp	303	12	1.50	-	0.22	-	0.16	-	[20]
MoOFOUR-Co-tpb	303	12	1.52	-	0.89	-	0.09	-	[21]
Mn-dhbq	303	12	1.81	-	1.80	0.93	-	-	[22]
CopzNi	303	12	2.13	1.37	-	-	0.07	-	[23]
HIAM-410	303	6.7	1.42	-	1.30	-	1.21	-	[24]

MFI	423	16	0.71	0.57	-	0.53	0.52	1.02	[25]
MFI	423	12	0.66	0.55	-	0.5	0.5	1.00	[25]
Zeolite BETA	423	16	0.82	-	0.81	0.74	0.7	1.16	[26]
Zeolite BETA	423	12	0.79	-	0.78	0.69	0.66	1.18	[26]
UU-200	303	12	1.67	1.23	1.12	0.37	0.1	3.70	[6]
Fe ₂ (BPD) ₃	433	16	-	1.12	1.32	1.1	1.09	1.01	[7]
Fe ₂ (BPD) ₃	433	12	1.28	1.04	1.16	0.92	0.92	1.00	[7]
Al-btoto	303	16	1.74	1.15	1.06	0.42	0.06	7.00	[8]
Al-btoto	303	12	1.71	1.13	1.04	0.41	0.05	8.20	[8]
ZU-62	298	12	1.71	0.87	0.62	0.15	0.11	1.36	This work
SIFSIX-2-Cu-i	298	12	1.98	1.11	1.05	0.25	0.14	1.79	This work
ZU-61	298	12				3.75	1.33	2.82	
ZU-72	298	16	2.12	1.19	1.23	0.26	0.15	1.73	This work
ZU-72	298	12	2	1.13	1.17	0.22	0.11	2.00	This work
SIFSIX-1-Cu	298	16	5.07		4.47	4.25	0.21	20.23	This work
SIFSIX-1-Cu	298	12	4.98		4.42	4.21	0.19	22.16	This work

5. *Concluding Remarks: Revise the conclusion to conform to the typical structure of summarizing key results, identifying unresolved issues, and presenting future perspectives. Any new information introduced should be incorporated into the results section, preserving the established structure for the conclusion.*

Response: We thank the reviewer for the valuable comment. As suggested, we have updated the conclusion in the revised manuscript and included the summarized key results, unresolved issues, and future perspectives.

Modification: Main text, page 10, line 252-265, added text “The construction and exploitation of the rotational freedom of organic linkers and SnF_6^{2-} anions within adaptive frameworks are efficient for the discrimination of alkane isomers with different degree of branching. The anion-functionalized ZU-72 realizes the complete size sieving of di-branched from mono-branched isomers with high selectivity and capacity. The framework with some degree of rotational freedom allows the rationally regulation of energy during adsorption and desorption processes, which contributes to the lower energy consumption of regeneration. And the successful separation of 22DMB and 23DMB by SIFSIX-1-Cu enables the further upgrading of gasoline to a maximum value (105). It should also be noted that to achieve efficient separations under practical operating conditions, the granulation of these materials and economic energy consumption assessments need to be considered, and we have already started to address these issues. Overall, this work offers a great advance towards gasoline upgrading, and also emphasizes the contribution of the responsive flexible skeleton, the intrinsic energy regulation makes the separation process be more efficient.”

6. *Minor Corrections:*

- *Define abbreviations like RON (Research Octane Number) when they are first mentioned.*
- *Ensure consistent temperature units (either K or °C) throughout the manuscript and accompanying figures.*
- *Address the identical nature of Figure 2b and Figure S13.*
- *In lines 108-111, include references to support the claims made.*

Response: We thank the reviewer for the careful review of this manuscript, and we have corrected the corresponding mistakes.

Modification: 1) Main text, page 1, abstract: added text “*Research Octane Number*”;

2) We have keep the consistent temperature units (K) throughout the manuscript and accompanying figures.

3) We have deleted the Figure S13 in the revised Supplementary Information.

4) The corresponding reference (*Angew. Chem. Int. Ed.* **62**, e202300722 (2023)) has been included in the revised manuscript.

REVIEWER COMMENTS

Reviewer #1 (Remarks to the Author):

All the previous concerns have been addressed.

Here comes a minor question: are those productivity values of high-quality gasoline consistent with their breakthrough curves including those in Figure 3? The claimed values of '15.8 mL g⁻¹' and '14.3 mL g⁻¹' seem far lower than those values indicated by horizontal axes.

Reviewer #2 (Remarks to the Author):

It can be accepted now.

Reviewer #3 (Remarks to the Author):

The authors have successfully addressed the reviewers comments in the revised version. Now, the paper can be published in Nature Communications.

Response to reviewers' comments

Comments from Reviewer 1:

All the previous concerns have been addressed. Here comes a minor question: are those productivity values of high-quality gasoline consistent with their breakthrough curves including those in Figure 3? The claimed values of '15.8 mL g⁻¹' and '14.3 mL g⁻¹' seem far lower than those values indicated by horizontal axes.

Response: We thank this reviewer for these positive and constructive comments about our work. In this work, the horizontal axis of the breakthrough curves represents the total volume of the mixture obtained by bubbling hexane isomers with nitrogen, rather than just the volume of hexane isomers, with nitrogen being the major component. For example, in the quinary mixture (nHEX/2MP/3MP/23DMB/22DMB) of Figure 3, the partial pressures of each component are 5.46 kPa, with the rest being nitrogen gas, therefore the calculated productivity values are lower than the values indicated by the horizontal axis.

We apologize for any confusion caused by the unclear writing, and have corrected the caption information of breakthrough curves in the revised manuscript and Supplementary Information, adding a note stating that the horizontal axis represents the volume of the mixture of nitrogen gas and hexane isomers.

Modification: The caption information of Figure 3 and Supplementary Figure 26, 27, 28, 29, 30, 31, 33, 34, 35 and 36 have been updated in the revised manuscript and Supplementary Information, adding a note stating that the horizontal axis represents the volume of the mixture of nitrogen gas and hexane isomers.

Comments from Reviewer 2:

It can be accepted now.

Response: We thank this reviewer for these positive comments about our work.

Comments from Reviewer 3:

The authors have successfully addressed the reviewers comments in the revised version. Now, the paper can be published in Nature Communications.

Response: We thank this reviewer for these positive comments about our work.

REVIEWERS' COMMENTS

Reviewer #1 (Remarks to the Author):

It can be published now.